# Disrupted miRNA Biogenesis Machinery Reveals Common Molecular Pathways and Diagnostic Potential in MDS and AML

**DOI:** 10.3390/biomedicines13123082

**Published:** 2025-12-14

**Authors:** Kenan Çevik, Mustafa Ertan Ay, Anıl Tombak, Özlem İzci Ay, Ümit Karakaş, Mehmet Emin Erdal

**Affiliations:** 1Department of Medical Biology and Genetics, Faculty of Medicine, Mersin University, 33343 Mersin, Türkiye; cvkknn@gmail.com (K.Ç.); ozlemay@mersin.edu.tr (Ö.İ.A.); mehmeteminerdal@gmail.com (M.E.E.); 2Department of Hematology, Faculty of Medicine, Mersin University, 33343 Mersin, Türkiye; aniltombak@mersin.edu.tr; 3Department of Pharmacy Services, Health Services Vocational School, Bayburt University, 69000 Bayburt, Türkiye; umitk.kas@gmail.com

**Keywords:** AML, MDS, miRNA biogenesis, *DROSHA/DICER1* genes dysregulation, precision diagnostics

## Abstract

**Background**: Myelodysplastic syndromes (MDS) and acute myeloid leukemia (AML) are clonal stem cell disorders in which disrupted post-transcriptional regulation contributes to aberrant hematopoiesis and leukemic transformation. The miRNA biogenesis machinery, which comprises Drosha, DGCR8, Dicer, TARBP2, and AGO1, ensures the precise maturation of miRNAs that control lineage commitment and proliferation. However, the extent to which alterations in this pathway reshape hematopoietic gene networks during myeloid disease evolution remains largely unexplored. **Methods**: Bone marrow samples from newly diagnosed, untreated MDS and AML patients and matched healthy controls were analyzed for the expression of five key miRNA biogenesis genes using quantitative real-time PCR. Statistical comparisons, correlation matrices, and ROC analyses were performed to characterize gene-expression differences. These results were integrated with multigene logistic modeling, decision-curve analysis, and exploratory random forest/SHAP approaches to evaluate molecular interactions and diagnostic relevance. **Results**: *DROSHA*, *DICER1*, and *TARBP2* were significantly downregulated in both MDS and AML, suggesting impaired miRNA maturation and a loss of global post-transcriptional control. *DGCR8* expression increased across higher-risk MDS groups, suggesting compensatory activation of the Microprocessor complex, whereas *AGO1* levels remained relatively stable, consistent with partial maintenance of RISC function. Correlation analyses revealed a co-regulated DROSHA–TARBP2–AGO1 module. ROC, logistic, and machine learning models identified *DGCR8* and *DICER1* as the strongest diagnostic discriminators. The integrated five-gene signature achieved high discriminative performance (AUC ≈ 0.98) and showed promise but remains preliminary potential for clinical application. **Conclusions**: Our findings suggest that defects in miRNA biogenesis disrupt hematopoietic homeostasis, reflecting common mechanisms in MDS and AML. The dysregulation of *DICER1*, *DGCR8*, and *TARBP2* offers insights into miRNA-driven leukemogenesis and may pave the way for miRNA-based diagnostic and therapeutic strategies, pending validation in larger cohorts. Although transcript-level data are provided, future studies should include functional validation to determine the impact on downstream miRNA processing and hematopoietic pathways.

## 1. Introduction

Acute myeloid leukemia (AML) and myelodysplastic syndromes (MDS) are clonal disorders of hematopoietic stem and progenitor cells characterized by ineffective hematopoiesis, impaired differentiation, and increased genomic instability. Large-scale genomic studies have identified recurrent mutations in *FLT3*, *NPM1*, *CEBPA*, *TP53*, and *SF3B1*, yet these alterations alone do not fully explain the marked clinical and biological heterogeneity of myeloid neoplasms [1,2,3]. In addition to these canonical drivers, recurrent mutations in epigenetic regulators such as *TET2, ASXL1*, and *DNMT3A*, and RNA splicing factors including *SF3B1*, *SRSF2,* and *U2AF1* reveal the involvement of multiple regulatory layers. These span chromatin remodeling to post-transcriptional control and collectively shape disease pathogenesis. Among these regulatory mechanisms, the disruption of microRNA (miRNA) biogenesis has recently emerged as a potential link between molecular lesions and disturbed hematopoietic differentiation.

miRNAs are short (~22 nucleotides), noncoding RNA molecules that fine-tune gene expression by guiding the degradation or translational repression of target mRNAs. The biosynthesis of miRNAs involves a series of well-coordinated enzymatic steps, collectively known as the miRNA biogenesis pathway. In the nucleus, Drosha and its cofactor DGCR8 form the microprocessor complex, which processes primary miRNA transcripts (pri-miRNAs) into precursor molecules (pre-miRNAs). These molecules are exported to the cytoplasm, where Dicer, assisted by its cofactor TARBP2, cleaves them into mature miRNA duplexes. Mature miRNAs are then incorporated into the RNA-induced silencing complex (RISC), where AGO1 mediates their interaction with target transcripts to regulate post-transcriptional gene silencing [4,5].

Disturbances in this pathway can profoundly alter miRNA abundance and function, thereby affecting key regulators of hematopoiesis. Drosha, an RNase III enzyme essential for pri-miRNA processing, initiates the nuclear phase of miRNA maturation. Decreased *DROSHA* expression has been detected in bone marrow stromal and hematopoietic cells from MDS patients, suggesting impaired production of tumor-suppressive miRNAs such as those of the miR-34 family under p53 control [6,7,8]. DGCR8, the double-stranded RNA binding partner of Drosha, determines the precision of pri-miRNA substrate recognition and cleavage. Its dysregulation can impair the microprocessor’s fidelity, weakening the p53-miRNA feedback loop and promoting aberrant progenitor expansion [9,10]. Beyond this regulatory axis, experimental models have shown that DGCR8 haploinsufficiency disrupts early stem cell differentiation and predisposes cells to myeloid transformation, underscoring its essential role in maintaining hematopoietic lineage balance [11].

The cytoplasmic enzyme Dicer finalizes miRNA maturation. Conditional deletion of *DICER1* in murine osteoprogenitors disrupts the bone marrow microenvironment and induces an MDS-like phenotype that can progress to secondary AML, underscoring the contribution of stromal miRNA processing to hematopoietic stability [8]. Similarly, reduced *DICER1* expression levels in patient’s MDS stromal cells correlate with cellular senescence and diminished support for hematopoietic stem cells [12].

TARBP2 (a Dicer-associated cofactor) stabilizes precursor miRNAs and enhances Dicer’s cleavage fidelity during the miRNA maturation process. A frameshift mutation identified in human cancers disrupts *TARBP2* translation, leading to defective miRNA processing and destabilization of Dicer [13]. Such impairment alters the global miRNA landscape and may influence cell differentiation and stress response pathways. Although data in hematologic malignancies remain limited, variations in *TARBP2* gene expression among AML subtypes could reflect lineage-specific differences in miRNA maturation efficiency.

AGO (Argonaute) proteins, the core effectors of the RISC, anchor mature miRNAs to their target transcripts and mediate post-transcriptional repression. Beyond this cytoplasmic role, AGO can also be associated with chromatin to modulate transcription in a small RNA-dependent manner. Experimental evidence has shown that chromatin bound AGO complexes participate in transcriptional regulation, providing a potential link between miRNA guided silencing and epigenetic control mechanisms [14]. Although not yet fully explored in hematologic malignancies, such nuclear functions may influence gene networks involved in hematopoietic differentiation and leukemic transformation.

Emerging data indicate that these biogenesis factors act in concert rather than in isolation. Cross-talk among Drosha, DGCR8, and Dicer likely determines overall processing fidelity and miRNA abundance. Hence, even modest changes in expression may produce cumulative disturbances in hematopoietic gene regulation and differentiation.

Despite such evidence, most existing studies have evaluated these genes individually or within cell-based systems that lack clinical context. Comprehensive assessments integrating multiple miRNA biogenesis genes in human MDS and AML cohorts remain scarce. This study therefore examined the expression of *DROSHA*, *DGCR8*, *DICER1*, *TARBP2*, and *AGO1* in bone marrow samples from untreated patients with MDS and AML, compared with healthy controls. Through combined expression, correlation, and diagnostic analyses, we aimed to determine whether concurrent alterations in the miRNA processing machinery might represent a shared molecular signature of myeloid transformation and possibly provide a framework linking defective biogenesis to the continuum between dysplasia and leukemic progression.

## 2. Materials and Methods

### 2.1. Patient Cohort and Diagnostic Classification

Bone marrow aspirates were collected from 20 newly diagnosed and previously untreated patients with acute myeloid leukemia (AML) and 34 patients newly diagnosed and previously untreated with myelodysplastic syndrome (MDS). A power analysis was performed prior to study initiation to ensure adequate statistical power. Although samples were obtained at initial presentation, final group assignment was made only after complete clinical, cytomorphologic, cytogenetic, and laboratory evaluation. Diagnostic confirmation and subclassification were performed by an experienced hematologist to ensure consistent and accurate classification.

The control group consisted of seven individuals without hematological or malignant disorders. Because bone marrow aspiration cannot ethically be performed in healthy volunteers, control samples were obtained from patients undergoing open-heart surgery, during which a minimal amount of marrow is incidentally aspirated from the sternal cavity and otherwise discarded. This ethically acceptable approach has been widely used in previous gene- and miRNA-expression studies. Only age and sex were available for the control group due to the intraoperative nature of sampling.

To minimize biological confounding across groups, strict exclusion criteria were applied during patient and control selection. All enrolled individuals (patients and controls) were required to be free of chronic systemic disease, active infection, prior chemotherapy or radiotherapy, regular medication use, smoking, alcohol consumption, or herbal supplement intake. These measures ensured that neither inflammatory conditions nor comorbidities influenced the gene-expression profiles analyzed.

AML diagnosis and subclassification were established according to the French–American–British (FAB) criteria, with the following distribution: AML-M0 (*n* = 7), AML-M1 (*n* = 5), AML-M2 (*n* = 1), AML-M4 (*n* = 4), and AML-M5 (*n* = 3). MDS classification followed the 2016 World Health Organization (WHO) criteria: MDS-SLD (*n* = 1), MDS-RS-SLD (*n* = 4), MDS-MLD (*n* = 16), MDS-EB1 (*n* = 10), and MDS-EB2 (*n* = 3). Risk stratification using the Revised International Prognostic Scoring System (IPSS-R) categorized patients into very low (*n* = 4), low (*n* = 12), intermediate (*n* = 14), high (*n* = 3), and very high (*n* = 1) risk groups.

All participants provided written informed consent, and the protocol was approved by the Institutional Clinical Research Ethics Committee (2015/227). The study was conducted in accordance with the Declaration of Helsinki (2013 revision). Comprehensive clinical and hematologic characteristics of the study population are summarized in Table 1.

### 2.2. RNA Extraction and cDNA Synthesis

Total RNA was extracted from bone marrow aspirate cells using TRIzol reagent (Invitrogen, Thermo Fisher Scientific, Waltham, MA, USA) according to the manufacturer’s protocol. RNA quantity and purity were determined spectrophotometrically using a NanoDrop 2000 (Invitrogen, Thermo Fisher Scientific, Waltham, MA, USA). Complementary DNA (cDNA) was synthesized from 2 µg of total RNA using Revertaid Reverse Transcriptase (Thermo Fisher Scientific, Vilnius, Lithuania). The reverse transcription reaction was carried out at 37 °C for 60 min, followed by enzyme inactivation at 95 °C for 5 min. The resulting cDNA samples were stored at −20 °C until further analysis.

### 2.3. Quantitative Real-Time PCR (qRT-PCR) Analysis

Quantitative real-time PCR for DROSHA, DGCR8, DICER1, TARBP2, and AGO1 genes was performed using the ABI Prism 7500 Real-Time PCR System (Applied Biosystems, Thermo Fisher Scientific, Waltham, MA, USA). Each 25 µL reaction mixture contained 12.5 µL of TaqMan Gene Expression Master Mix, 5 µL of cDNA, 2.5 µL of each primer, 0.6 µL of probe, and nuclease-free water. Thermal cycling conditions were as follows: 50 °C for 2 min, 95 °C for 10 min, followed by 50 cycles of 95 °C for 15 s and 60 °C for 1 min.

Primer and probe sequences specific to *DROSHA*, *DGCR8*, *DICER1*, *TARBP2*, *AGO1*, and *β-actin* were designed using Primer Express 3.0 (Applied Biosystems, Thermo Fisher Scientific, Waltham, MA, USA) and are listed in Table 2. In probe design, cytosine residues were substituted with *C-5-propynyl-dC (pdC)* to enhance hybridization affinity and duplex stability (increasing Tm by approximately 2–8 °C per substitution). Candidate oligonucleotides were evaluated through NCBI BLAST (BLAST+; National Center for Biotechnology Information, Bethesda, MD, USA) alignment to ensure target specificity, avoidance of SNP-containing regions, and compatibility across transcript isoforms. Only sequences demonstrating high theoretical specificity were synthesized. All assays were experimentally validated using Total RNA Control (Applied Biosystems, Cat. No. 4307281), and any primer–probe set failing to produce reproducible amplification was discarded. The assays for *DROSHA*, *DGCR8*, and *DICER1* had also been validated previously in an independent peer-reviewed study [15], demonstrating robust amplification performance in a separate patient cohort. In the present dataset, all five assays generated clear exponential amplification curves with consistent Ct values across biological replicates, and no-template controls remained negative. The use of TaqMan chemistry with pdC-modified probes further ensured high hybridization specificity and reliable amplification efficiency within the tested dynamic range.

While *β-actin* was selected as the reference gene based on previous hematologic studies, possible variability within bone marrow tissue is acknowledged as a methodological limitation of the study.

RNA integrity and purity were assessed using A260/280 ratios between 1.8 and 2.0. Relative gene expression was calculated using the 2^−ΔΔCt^ method. All qPCR reactions were performed in duplicate, and specificity was ensured by the absence of nonspecific amplification in negative controls and by uniform amplification kinetics across samples.

### 2.4. Statistical Analysis

All statistical analyses were performed using GraphPad Prism 10.0 (GraphPad Software, San Diego, CA, USA) and Statistica 13.3.1 (StatSoft, Tulsa, OK, USA). Gene expression data were analyzed using the 2^−ΔΔCt^ method, and results were expressed as relative fold changes. Data distribution was evaluated using the Shapiro–Wilk test to determine normality. Because most variables did not meet normal distribution assumptions, nonparametric tests were applied. Comparisons of miRNA biogenesis gene expression across AML, MDS, and control groups were conducted using the Kruskal–Wallis test, followed by pairwise Mann–Whitney U tests with Bonferroni adjustment for multiple comparisons. Given the limited number of targeted biomarkers, no additional multiple-testing correction (e.g., FDR) was applied beyond the Bonferroni adjustment, which is acknowledged as a statistical limitation. Associations between gene expression and clinicopathologic features (WHO subtypes, IPSS-R risk scores, and FAB categories) were assessed using the Kruskal–Wallis or Mann- Whitney tests as appropriate.

Intergene relationships were explored using Spearman’s rank correlation analysis, and results were visualized in correlation heatmaps to highlight co-regulatory patterns among biogenesis components. To evaluate diagnostic performance, receiver operating characteristic (ROC) curve analyses were performed for each gene using bootstrap resampling (1000 iterations) to estimate 95% confidence intervals (CI) for the area under the curve (AUC). Youden’s J index was used to determine optimal cutoff values and corresponding sensitivity and specificity.

A multigene logistic regression model incorporating all five genes was constructed to assess integrated diagnostic performance. Model discrimination was evaluated using AUC values, and comparisons with the best-performing single gene were tested using DeLong’s test. Calibration quality was assessed by Brier scores and the Hosmer-Lemeshow goodness-of-fit test, while visual calibration curves examined agreement between predicted and observed probabilities. Decision Curve Analysis (DCA) was performed to estimate the net clinical benefit across a range of threshold probabilities, providing insight into the potential clinical utility of the multigene model. All tests were two-tailed, and a *p*-value < 0.05 was considered statistically significant.

Additionally, an exploratory random forest model with SHAP interpretation was implemented for MDS and AML versus controls to assess the gene-level diagnostic impact. Models were trained on z score standardized expression data using 5-fold stratified cross validation (AUC as metric). Findings were treated as hypothesis-generating due to the limited control sample size.

## 3. Results

### 3.1. Expression Profiles of miRNA Biogenesis Genes in AML, MDS, and Control Groups

Relative expression levels of the key miRNA biogenesis genes (*DICER1*, *DROSHA*, *DGCR8*, *TARBP2*, and *AGO1*) were evaluated across AML, MDS, and control samples using the 2^−ΔΔCt^ method. The comparative distribution of these genes is illustrated in Figure 1, and the statistical results of Kruskal–Wallis and pairwise Mann–Whitney U analyses are summarized in Table 3.

Overall, distinct expression patterns were observed among the three groups. *DICER1* and *DROSHA* transcripts showed a consistent and statistically significant reduction in both AML and MDS compared with controls (*p* < 0.05), suggesting global downregulation of the early miRNA-processing machinery in myeloid malignancies. Similarly, TARBP2, a cofactor essential for Dicer-mediated pre-miRNA cleavage, exhibited a marked decline in AML and MDS relative to controls (*p* < 0.01), reinforcing the notion of coordinated impairment in cytoplasmic miRNA maturation.

In contrast, *DGCR8* expression displayed a divergent trend. While MDS samples showed relatively lower *DGCR8* expression compared with controls, AML cases exhibited considerable inter-individual variability, with a subset demonstrating increased levels (*p* < 0.01). This heterogeneity may indicate disease stage dependent deregulation of the Microprocessor complex, potentially linked to altered pri-miRNA recognition or feedback loops between DGCR8 and DROSHA.

No statistically significant difference was detected in *AGO1* expression among the three groups, implying that downstream AGO1-mediated RNA silencing is less affected in the early and late stages of myeloid transformation.

Collectively, these results demonstrate that aberrant regulation of the Dicer/Drosha/DGCR8 axis constitutes a shared molecular feature of MDS and AML, while the persistence of *AGO1* expression may reflect a partial retention of post-transcriptional silencing capacity. These findings highlight a disrupted miRNA-processing network that could contribute to defective hematopoietic differentiation and malignant progression.

### 3.2. Clinical Associations of miRNA Biogenesis Gene Expression

The expression levels of *DROSHA*, *DGCR8*, *DICER1*, *TARBP2*, and *AGO1* were evaluated in relation to major clinicopathologic variables in both the MDS and AML cohorts.

In the MDS group, gene expression levels were first compared across WHO subtypes (MDS-SLD, MDS-RS-SLD, MDS-MLD, MDS-EB1, and MDS-EB-2); however, no statistically significant differences were observed (*p* > 0.05 for all genes). Similarly, expression profiles analyzed according to IPSS-R risk categories revealed no significant variation for *DROSHA*, *DICER1*, *TARBP2*, or *AGO1*. In contrast, *DGCR8* expression showed a significant stepwise increase with higher IPSS-R risk levels (*p* = 0.008, Figure 2A), suggesting that enhanced activity of the nuclear Microprocessor complex may accompany disease progression in MDS. No significant associations were found between gene expression and bone marrow blast percentage or number of cytopenias (all *p* > 0.05).

In the AML cohort, miRNA biogenesis gene expression was assessed according to FAB subtypes (M0–M6). Among the analyzed genes, only *TARBP2* showed a statistically significant difference across FAB categories (*p* = 0.022; Figure 2B), with a tendency toward lower expression in more differentiated subtypes. Expression levels of *DROSHA*, *DGCR8*, *DICER1*, and *AGO1* did not differ significantly across FAB classes (all *p* > 0.05).

These findings indicate that while most miRNA biogenesis genes exhibit stable expression across clinical subgroups, *DGCR8* in MDS and *TARBP2* in AML display disease-related transcriptional alterations that may be linked to disease severity or differentiation status.

### 3.3. Correlation Analysis Among miRNA Biogenesis Genes

Spearman correlation analysis, performed on all samples (MDS, AML, and controls combined), revealed strong positive correlations among *DROSHA, AGO1*, and *TARBP2* (r = 0.44–0.82, *p* < 0.001), indicating coordinated regulatory behavior within the downstream steps of miRNA biogenesis. In contrast, DGCR8 showed moderate inverse correlations with *DROSHA* and *DICER1* (r ≈ −0.25 to −0.71, *p* < 0.05), suggesting a compensatory mechanism within the Microprocessor complex. Figure 3 illustrates the complete correlation structure, showing a distinct cluster of positively correlated genes centered around *DROSHA* and *AGO1.* These associations imply that DROSHA/AGO1/TARBP2 function as a co-regulated module that supports mature miRNA formation, while DGCR8 may act through feedback control to maintain equilibrium in the biogenesis machinery.

### 3.4. Diagnostic Performance of miRNA Biogenesis Genes

To evaluate the diagnostic value of the five miRNA-biogenesis genes, ROC curve analyses were performed for each gene in comparisons between disease groups and healthy controls. The AUC values with 95% bootstrap confidence intervals, thresholds, and corresponding sensitivities and specificities are presented in Table 4, and representative ROC plots are shown in Figure 4, Figure 5 and Figure 6. Bootstrap resampling (1000 iterations) confirmed the stability of AUC estimates, producing narrow 95% confidence intervals for all genes.

#### 3.4.1. MDS and Control Comparison

All analyzed genes showed significant discriminatory ability between MDS patients and healthy controls. Among them, *DGCR8* exhibited the highest diagnostic accuracy (AUC = 0.960, *p* < 0.0001), followed by *DICER1* (AUC = 0.866) and *TARBP2* (AUC = 0.819). *DROSHA* and *AGO1* also demonstrated moderate but statistically significant diagnostic power (AUC = 0.777 and 0.790, respectively). The curves indicate that *DGCR8* overexpression and the downregulation of other genes can effectively distinguish MDS cases from healthy controls (Figure 4). The strong diagnostic performance of *DGCR8* and *DICER1* supports their potential as early molecular indicators of disturbed miRNA biogenesis in myelodysplasia.

#### 3.4.2. AML and Control Comparison

Similarly, all five genes displayed substantial diagnostic value for differentiating AML patients from controls. The highest AUCs were observed for *DGCR8* (AUC = 0.971) and *DICER1* (AUC = 0.950), while *TARBP2*, *DROSHA*, and *AGO1* showed slightly lower but still significant accuracies (AUCs = 0.875, 0.886, and 0.836, respectively). These ROC patterns reflect a consistent imbalance in miRNA processing genes across both MDS and AML (Figure 5). The parallel diagnostic power of *DGCR8* and *DICER1* in AML suggests that perturbations in early miRNA processing may accompany malignant transformation of dysplastic clones.

#### 3.4.3. AML and MDS Comparison

To investigate whether these genes could discriminate between AML and MDS, an additional ROC analysis was performed. None of the five genes demonstrated meaningful discriminative ability between the two diseases, with AUC values close to 0.5 for *DGCR8*, *DICER1*, *DROSHA*, and *AGO1*, and a slightly lower AUC for *TARBP2* (AUC = 0.296). This finding indicates that while expression profiles distinguish patients from controls, they do not clearly separate AML from MDS (Figure 6). The absence of clear diagnostic separation supports the view that dysregulation of miRNA biogenesis represents a shared molecular feature of both disorders rather than a stage specific marker.

### 3.5. Multigene Logistic Model and Calibration Performance

The calibration performance of the multigene logistic regression model was evaluated to assess the agreement between predicted probabilities and observed outcomes. As shown in Figure 7, both MDS and AML models demonstrated good calibration, with data points lying close to the ideal 45° line. This indicates that the predicted probabilities generated by the five-gene model are well aligned with the actual frequency of disease, supporting the reliability of this logistic model across probability ranges.

Overall, ROC analyses highlight *DGCR8* and *DICER1* as reliable diagnostic indicators of myeloid neoplasia, although their expression fails to discriminate between disease stages.

### 3.6. Model Validation and Clinical Utility Analysis

To further evaluate the diagnostic robustness of the integrated model, a multigene logistic regression analysis was performed incorporating *DROSHA*, *DGCR8*, *DICER1*, *TARBP2*, and *AGO1*. Compared to the best-performing single gene, the multigene model achieved higher overall discriminative power in both MDS and AML cohorts. The model yielded an AUC of 0.978 for MDS and 0.982 for AML, indicating excellent diagnostic accuracy. When compared with the top single gene (*DGCR8*), DeLong’s test confirmed a statistically significant improvement in predictive performance (MDS: *p* < 0.001; AML: *p* = 0.004), supporting the additive effect of combining multiple miRNA-biogenesis genes (Table 5, Figure 8A,B).

The Brier score and Hosmer-Lemeshow (HL) test were applied to assess the model’s calibration accuracy. Both cohorts showed low Brier scores (MDS: 0.08; AML: 0.07) and non-significant HL *p*-values (MDS: *p* = 0.42; AML: *p* = 0.38), demonstrating good agreement between predicted probabilities and observed outcomes (Table 5). To explore the potential clinical applicability, Decision Curve Analysis (DCA) was performed. As shown in Figure 9A,B, the multigene model consistently provided a higher net clinical benefit across a broad range of threshold probabilities compared with “treat-all” and “treat-none” strategies.

This suggests that integrating these five miRNA biogenesis genes into a composite model may offer superior clinical utility and better patient stratification than any single gene-based approach. Overall, the combined model not only improved diagnostic accuracy but also demonstrated promising translational potential for future clinical implementation.

### 3.7. Random Forest and SHAP (SHapley Additive exPlanations) Analysis

To complement conventional statistical analyses, an exploratory machine learning approach was performed using a random forest classifier with SHAP interpretation to further evaluate the individual diagnostic contribution of miRNA biogenesis genes. Separate models were developed for MDS versus controls and AML versus controls using the expression levels of *DROSHA*, *DGCR8*, *DICER1*, *TARBP2*, and *AGO1* as predictors. Gene expression data were standardized and model performance was assessed using 5-fold stratified cross-validation, with AUC as the primary evaluation metric.

The models showed high discriminative performance (AUC = 0.94 for MDS and 1.00 for AML). In the MDS model, *DGCR8* was the most influential classifier component, followed by *TARBP2* and *DICER1*, whereas *AGO1* contributed minimally. In the AML model, *DGCR8* and *DICER1* consistently ranked as the strongest predictors, with *DROSHA* and *TARBP2* having moderate impact. These machine learning findings were consistent with the ROC analysis and further confirmed by SHAP dependence plots, which demonstrated positive shifts in model output with increased *DGCR8* (MDS) or *DICER1* (AML) expression (Appendix A).

Given the limited number of controls and potential risk of model overfitting, these results are considered exploratory and hypothesis-generating and should be interpreted with caution.

## 4. Discussion

Our study demonstrates significant disruptions in several key miRNA biogenesis genes (*DROSHA*, *DGCR8*, *DICER1*, *TARBP2*, and *AGO1*) in MDS and AML. The coordinated dysfunction of these genes points to a broader defect in miRNA maturation, contributing to impaired hematopoiesis and leukemogenesis. This aligns with a growing body of evidence suggesting that myeloid malignancies arise from multi-layered disruptions in cellular processes, including the deregulation of miRNA biogenesis and the loss of miRNA-dependent post-transcriptional control over hematopoietic differentiation [1,2,3].

The reduced expression of *DROSHA* and *DICER1* in MDS and AML is consistent with their established roles in initiating pri-miRNA cleavage and processing, respectively. As both enzymes are essential for miRNA maturation, their downregulation leads to global miRNA depletion, disrupting tumor suppressive pathways and favors clonal expansion. Consistent with previous reports of their association with poor prognosis in solid tumors [16], our findings extend the relevance of *DICER1* and *DROSHA* alterations to hematologic malignancies as well. Our study further expands this understanding by showing that dysregulation of *DICER1* and *DROSHA* is a common feature in hematologic malignancies, not only in myeloid neoplasms but also in lymphoid disorders such as Hodgkin lymphoma [17]. Additionally, *DICER1* deficiency in stromal cells has been shown to contribute to premature senescence and diminished support for hematopoietic stem cells, which may further impair the marrow niche and enhance the leukemic potential of the disease [7,18]. Moreover, pan-cancer studies have identified significant variations in the expression of miRNA-processing genes, including *DICER1*, *DGCR8*, and *TARBP2*, across multiple malignancies, reinforcing the importance of miRNA biogenesis defects in neoplastic progression [19].

Interestingly, *DGCR8* expression showed a progressive increase across higher-risk IPSS-R categories in MDS patients, suggesting a potential compensatory activation of the miRNA biogenesis pathway in response to disease progression. Consistent with reports linking DGCR8 activity to stem cell self-renewal and malignant transformation [9,10] and its association with stemness, DNA damage tolerance, and drug resistance phenotypes [20], its progressive upregulation in higher-risk MDS may reflect an adaptive mechanism aimed at sustaining minimal miRNA processing capacity under oncogenic stress, thereby supporting the persistence of undifferentiated progenitor cells. This pattern may represent an adaptive mechanism to maintain miRNA processing activity under oncogenic stress; however, this hypothesis requires confirmation through experimental studies.

TARBP2 downregulation, particularly in the AML subtypes, further supports our hypothesis that miRNA biogenesis defects contribute to leukemogenesis. TARBP2 stabilizes Dicer, and its loss has been linked to impaired miRNA maturation and increased miRNA turnover. Recent studies have shown that TARBP2 plays a pivotal role in controlling miRNA levels by interacting with both miRNA processing complexes and exoribonucleases, which is critical for maintaining cellular homeostasis [21]. While early claims of *TARBP2* mutations in cancers were later retracted [22], our results point to TARBP2 deficiency as a major contributor to the impaired miRNA landscape observed in myeloid malignancies.

Unlike the upstream processing components, *AGO1* expression remained relatively stable among our patient groups. AGO1, a core component of the RISC, is essential for miRNA-mediated gene silencing. Its stable expression of *AGO1* may suggest partial preservation of miRNA-guided repression despite overall miRNA depletion. Interestingly, AGO1 is also involved in chromatin associated RNA interference, which links post-transcriptional regulation to transcriptional control and epigenetic remodeling [14]. In hematologic malignancies, alterations in AGO1 function could potentially contribute to the dysregulation of hematopoietic gene expression by influencing chromatin architecture and transcriptional reprogramming, thus contributing to leukemic transformation [21]. Correlation analysis further supported a coordinated behavior among DROSHA, TARBP2, and AGO1, indicating a partially preserved cytoplasmic processing module despite upstream disruptions in DROSHA and Dicer. However, these correlations represent associative patterns only and should not be interpreted as evidence of direct causal relationships between gene dysregulation and disease status. Its stable expression may reflect partial preservation of RISC activity despite upstream impairment, although this interpretation remains speculative without functional validation.

Despite these molecular insights, the impact of *AGO1* gene deregulation in hematologic cancers is not fully understood. While *AGO1* mutations have not been conclusively identified in MDS or AML, its dysregulated silencing could disturb gene expression networks critical for hematopoietic differentiation and leukemogenesis, further highlighting the complexity of miRNA regulation in malignancies.

Our study also highlights the utility of miRNA biogenesis gene expression as a diagnostic tool. *DICER1* and *DGCR8* emerged as key biomarkers, with strong diagnostic discrimination between patients and controls. The ROC analyses revealed high sensitivity against controls, indicating that these genes may serve as useful biomarkers for distinguishing patients from healthy individuals; however, they do not demonstrate discriminatory power between AML and MDS [23]. The multigene model integrating all five genes significantly improved both diagnostic power and calibration, reinforcing the potential value of a multi-gene signature for assessing disease risk. However, consistent with our ROC analyses, none of the analyzed genes clearly differentiated AML from MDS patients, suggesting that defective miRNA biogenesis represents a shared molecular hallmark rather than a disease stage specific alteration. This multi-gene signature may complement traditional cytogenetic and morphologic diagnostic criteria, offering a more precise risk stratification tool for clinical practice. Taken together, the progressive increase in *DGCR8* expression in high-risk MDS may reflect an adaptive response that enables malignant progenitor cells to maintain minimal miRNA-processing capacity, supporting clonal persistence despite upstream processing defects. Such compensatory activation may contribute to progression toward more aggressive disease phenotypes. Our exploratory machine learning analyses (random forest and SHAP) were consistent with these observations, highlighting *DGCR8* and *DICER1* as the strongest contributors to diagnostic separation, with *DGCR8* remaining the leading classifier in MDS. Although these trends support the robustness of the observed expression patterns observed, gene-specific variability and the limited cohort size indicate that these results should be interpreted with caution. External validation in independent cohorts will be essential to confirm the diagnostic generalizability of this multi-gene signature.

Supporting these findings, both random forest and SHAP analyses further confirmed the dominant diagnostic contribution of *DGCR8* and *DICER1* in AML, while in MDS, *DGCR8* remained the leading contributor, followed by *TARBP2* and *DICER1* at moderate levels (Appendix A). Although AUC values were high, gene-specific variations in sensitivity and specificity were observed, likely reflecting molecular heterogeneity and the limited sample size. These findings were further supported by machine learning analyses; however, external validation is required to confirm diagnostic generalizability. Therefore, the conclusions regarding clinical applicability should be considered preliminary. Prospective validation in larger, independent cohorts will be necessary to assess real-world diagnostic performance and reproducibility of the five-gene signature.

Looking forward, miRNA-based therapies targeting upstream steps of miRNA processing—such as Dicer and DGCR8—may offer novel therapeutic approaches for hematologic malignancies characterized by impaired miRNA biogenesis. Recent preclinical studies have shown promising results in restoring Dicer activity or supplementing tumor- suppressive miRNAs in cancer models [17,24]. These strategies could support precision-medicine approaches tailored to patients with defective miRNA-processing machinery. Additionally, large language models (LLMs) are increasingly being explored in biomedical sciences and may eventually assist in interpreting complex molecular profiles. Nonetheless, their use in hematologic disease diagnostics remains experimental and requires validation in larger datasets before clinical integration. Future studies should also explore whether combining mature miRNA-expression profiles with biogenesis-gene signatures enhances diagnostic or prognostic accuracy.

Our study is limited by its cross-sectional design, which does not allow for longitudinal monitoring of miRNA-biogenesis changes over time. Longitudinal follow-up from MDS to AML would help clarify how alterations in miRNA processing contribute to disease progression. Furthermore, the use of total RNA extraction from bone marrow samples does not capture cell-type-specific regulatory dynamics, and single-cell analysis could provide deeper biological insight. In particular, the limited size of the control group may have affected the robustness of ROC estimates and should be interpreted with caution. As this was a transcript-level expression study without functional validation, we could not directly determine whether dysregulation of *DROSHA*, *DICER1*, or *TARBP2* alters hematopoietic differentiation, miRNA processing efficiency, or downstream signaling pathways. Functional assays (e.g., knockdown or overexpression models) will therefore be important in future studies to validate the pathogenic impact of these key biogenesis genes. Moreover, the study lacks an independent external validation cohort, which limits the generalizability of the observed diagnostic performance. Although the findings are promising, the absence of external validation means that we cannot confidently extrapolate these results to broader patient populations. Validation in larger, externally recruited datasets will therefore be essential to confirm reproducibility and clinical applicability.

Future investigations integrating small-RNA sequencing, functional-validation assays, and external validation in independent cohorts will be crucial to confirm these mechanistic implications and provide a more comprehensive understanding of the regulatory consequences of impaired miRNA biogenesis. Additionally, future studies should aim to replicate these findings in independent cohorts and further investigate the diagnostic performance of the identified biomarkers in diverse clinical settings. Adjustment for potential clinical confounders (e.g., age, cytogenetic abnormalities, comorbidities) was restricted by data availability and should be considered when interpreting the diagnostic results.

## 5. Conclusions

Our findings demonstrate that dysregulation of key miRNA biogenesis genes, particularly *DROSHA*, *DICER1*, *DGCR8*, and *TARBP2*, is associated with hematopoietic disruption in newly diagnosed MDS and AML patients. The coordinated alterations observed suggest that impaired miRNA processing may contribute to leukemogenesis through disruption of post-transcriptional regulatory pathways. While *DGCR8* and *DICER1* emerged as promising diagnostic indicators, particularly in AML, the observed variability in gene-specific sensitivity and specificity, together with the limited size of the control cohort, indicates that these results should be interpreted with caution. The multigene expression model provided enhanced discriminative capacity compared to single genes, supporting the potential utility of miRNA biogenesis signatures in refinement of diagnostic assessment. However, due to the absence of functional validation and external cohort testing, the clinical applicability of this signature remains preliminary. As noted in the Discussion, the lack of an external validation cohort is a key limitation, and replication in independent, larger cohorts remains essential to confirm the diagnostic performance and generalizability of our findings.

Furthermore, although this study provides valuable insights into miRNA-biogenesis disruption in myeloid malignancies, the absence of functional assays limits our ability to directly assess the impact of these changes on downstream hematopoietic pathways. Future investigations integrating functional assays, small-RNA sequencing, and validation in independent, larger cohorts will be critical to establish the clinical value of these biogenesis-related molecular markers. If confirmed, such insights may support the development of targeted therapeutic strategies aimed at restoring miRNA-processing fidelity in hematologic malignancies.

## Figures and Tables

**Figure 1 biomedicines-13-03082-f001:**
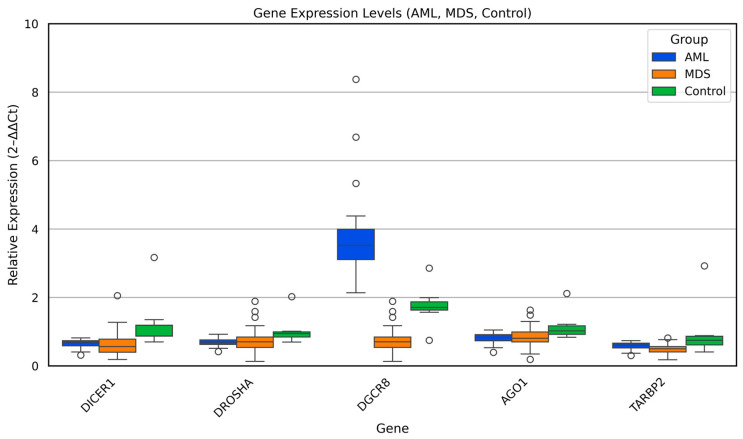
Relative expression of miRNA biogenesis genes *(DICER1, DROSHA, DGCR8, TARBP2,* and *AGO1*) across AML, MDS, and control groups. Box plots show median and interquartile ranges of 2^−ΔΔCt^ values.

**Figure 2 biomedicines-13-03082-f002:**
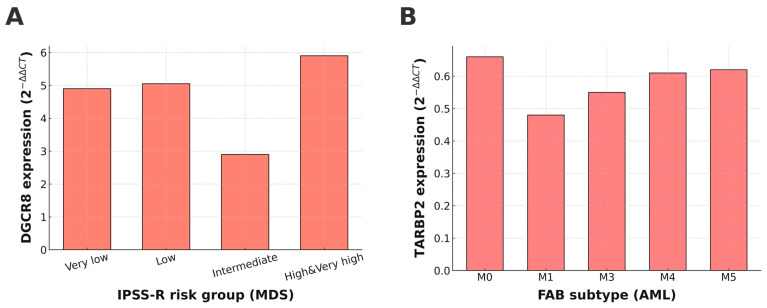
Expression patterns of *DGCR8* and *TARBP2* across clinical subgroups of MDS and AML. (**A**) *DGCR8* expression showed a significant stepwise increase with higher IPSS-R risk categories in MDS (*p* = 0.008). (**B**) *TARBP2* expression varied significantly among FAB subtypes in AML (*p* = 0.022).

**Figure 3 biomedicines-13-03082-f003:**
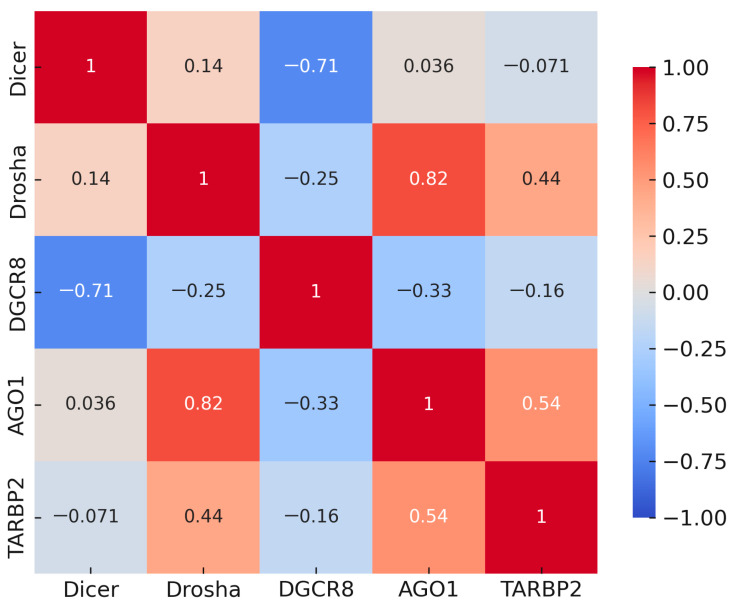
Spearman correlation matrix of *DROSHA*, *DICER1*, *DGCR8*, *AGO1,* and *TARBP2* across all samples (MDS, AML, and controls combined). Red indicates positive and blue indicates negative correlations; clustering among *DROSHA-AGO1-TARBP2* contrasts with inverse associations of *DGCR8* with *DROSHA/DICER1*.

**Figure 4 biomedicines-13-03082-f004:**
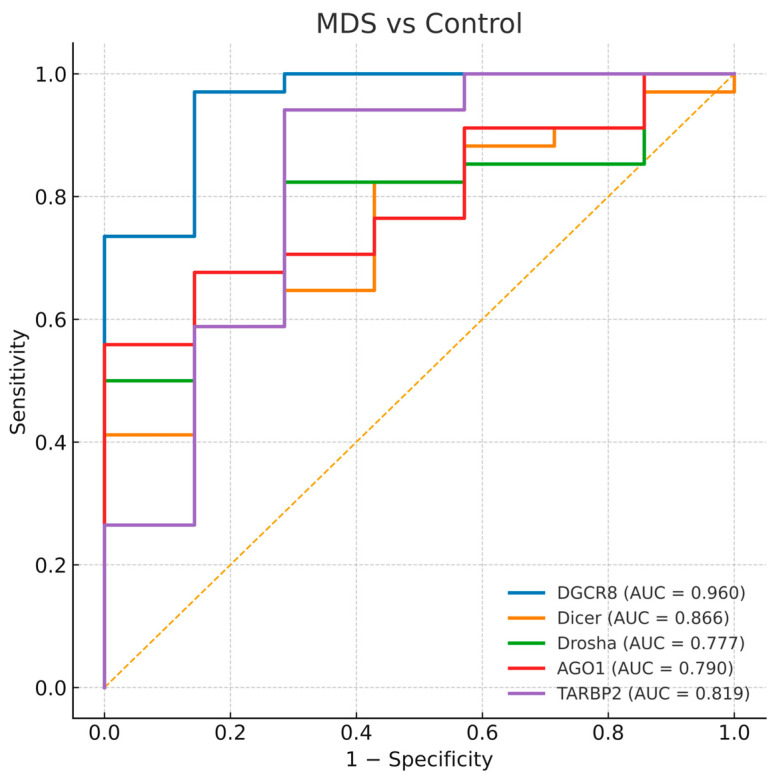
Receiver operating characteristic (ROC) curves for *DGCR8*, *DICER1*, *DROSHA*, *AGO1*, and *TARBP2* distinguishing MDS-controls. The diagonal dashed line represents the reference line indicating no discriminative ability (AUC = 0.5).

**Figure 5 biomedicines-13-03082-f005:**
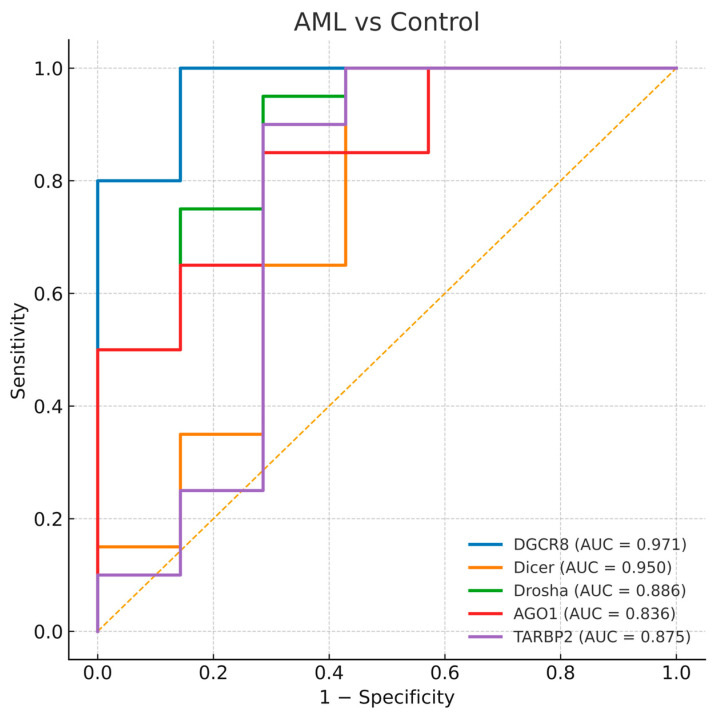
Receiver operating characteristic (ROC) curves for *DGCR8*, *DICER1*, *DROSHA*, *AGO1*, and *TARBP2* distinguishing AML-controls. The diagonal dashed line represents the reference line indicating no discriminative ability (AUC = 0.5).

**Figure 6 biomedicines-13-03082-f006:**
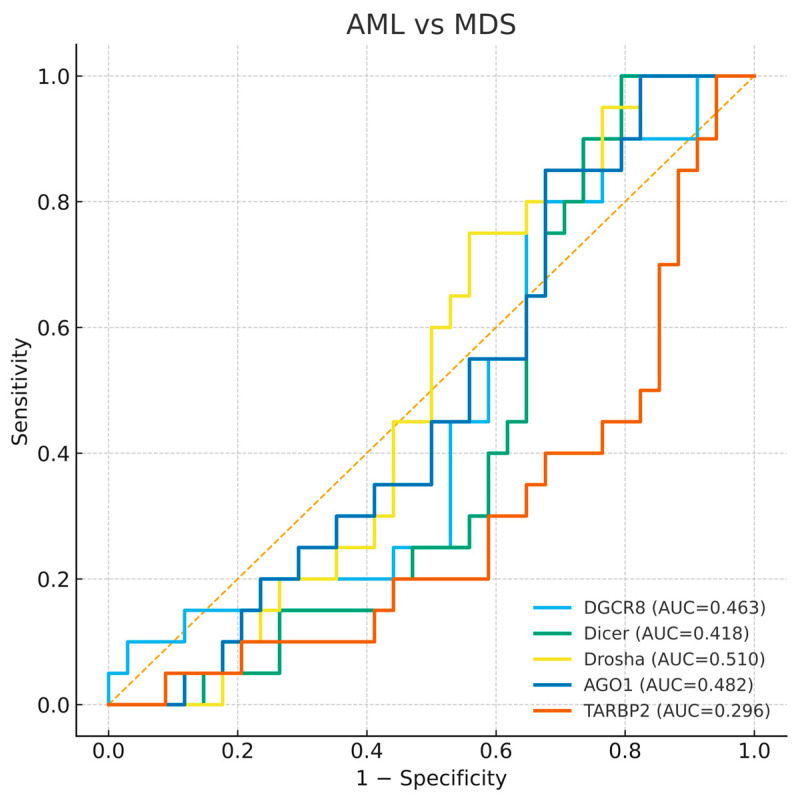
Receiver operating characteristic (ROC) curves for *DGCR8*, *DICER1*, *DROSHA*, *AGO1*, and *TARBP2* distinguishing AML-MDS. The diagonal dashed line represents the reference line indicating no discriminative ability (AUC = 0.5).

**Figure 7 biomedicines-13-03082-f007:**
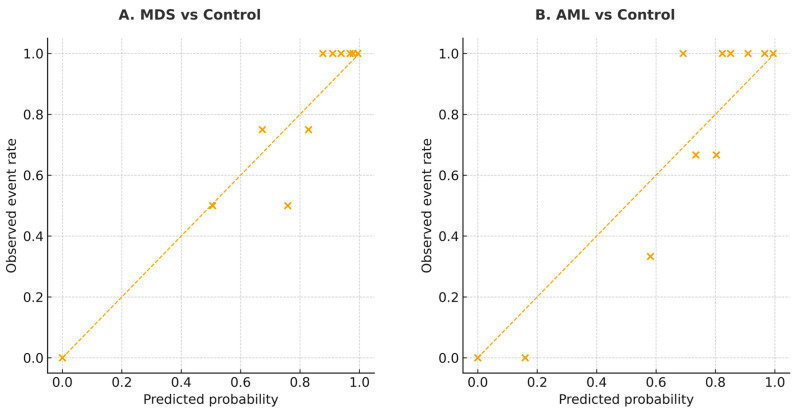
Calibration plots for the multigene logistic model in MDS and AML. (**A**) MDS vs. Control. (**B**) AML vs. Control. The diagonal line indicates perfect calibration between predicted probabilities and observed event rates. The diagonal line indicates perfect calibration. Each “×” symbol represents the observed event rate plotted against the corresponding predicted probability for each risk group.

**Figure 8 biomedicines-13-03082-f008:**
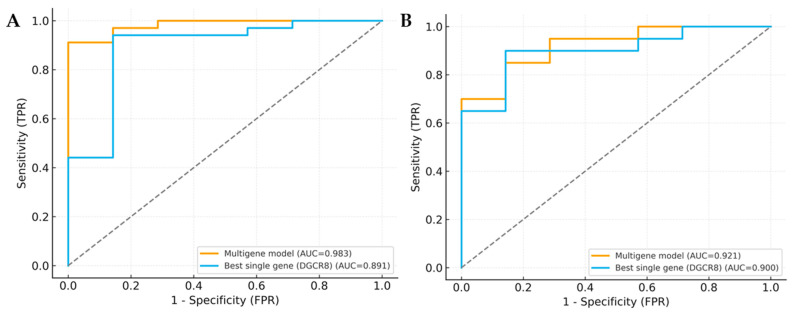
ROC curve comparison of multigene model and single gene. (**A**) ROC curves for MDS-control. (**B**) ROC curves for AML-control. Curves compare the multigene logistic model (*DROSHA*, *DGCR8*, *DICER1*, *TARBP2*, and *AGO1*) with the best single gene (*DGCR8*). Gray dotted lines indicate gridlines added for visual guidance.

**Figure 9 biomedicines-13-03082-f009:**
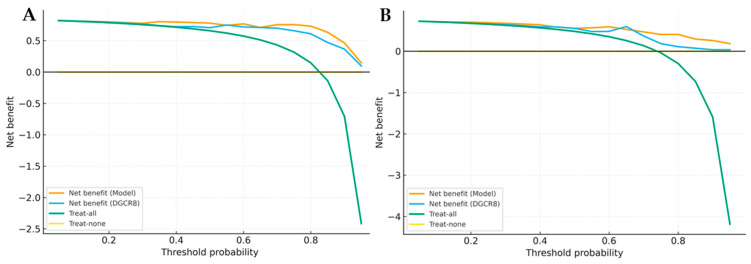
Decision curve analysis (DCA) of multigene model and single gene. (**A**) DCA for MDS versus control. (**B**) DCA for AML versus control. Curves show the clinical net benefit of the multigene model compared with the best single gene (*DGCR8*) and reference strategies (“treat-all”, “treat-none”).

**Table 1 biomedicines-13-03082-t001:** Clinical and some hematologic characteristics of study groups.

Characteristic	Controls (*n* = 7)	MDS (*n* = 34)	AML (*n* = 20)
**Gender (M/F)**	5/2	19/15	10/10
**Age years** **(median, range)**	64 (55–76)	69 (50–88)	63 (43–77)
**WHO 2016** **classification**	-	MDS-SLD: 1 MDS-RS-SLD: 4 MDS-MLD: 16 MDS-EB1: 10 MDS-EB2: 3	-
**IPSS-R category**	-	Very low: 4 Low: 12 Intermediate: 14 High/Very high: 4	-
**Number of cytopenias**	-	0: 5; 1: 14; 2: 13; 3: 2	-
**BM blast percentage**	-	<5%: 21 ≥5–<10%: 10 ≥10–<29%: 3	-
**FAB classification**	-	-	M0: 7, M1: 5, M2: 1, M4: 4, M5: 3
**Cytogenetic risk**	-	-	Good: 3; Intermediate: 14; Poor: 3

**Table 2 biomedicines-13-03082-t002:** Primer and probe sequences of miRNA biogenesis genes used in qRT-PCR.

Gene/ID	Primers and Probe Sequences
** *DROSHA* ** **29102 ***	F 5′-GAACAGTTCAACCCCGATGTG-3′ R 5′-CTCAACTGTGCAGGGCGTATC-3′ PR 5′-FAM-TTA(pdC)TTTT(pdC)CGATTAT(pdC)GTC-ZNA4-BHQ-1-3′
** *DGCR8* ** **54487 ***	F 5′-TCTTTGAATGTGAGAACCCAAGTG-3′ R 5′-CCGTAAGTCACACCATCAATGG-3′ PR 5′-FAM-CCTTTTGGTGCCTCGGT-ZNA4-BHQ-1-3′
** *DICER1* ** **23405 ***	F 5′-CCCGGCTGAGAGAACTTACG-3′ R 5′-TGTAACTTCGACCAACACCTTTAAAT-3′ PR 5′-FAM-CGGGAAGGT(pdC)AGAGT(pdC)A-ZNA4-BHQ-1-3′
** *TARBP2* ** **26895 ***	F 5′-GAAGGCAGCCAAGCACAAG-3′ R 5′-CTCCCCCCTTTGAGGTGTTT-3′ PR 5′-FAM-CAGCTGAGGTGGCCCTC-ZNA4-BHQ-1-3′
** *AGO1* ** **1126523 ***	F 5′-CAGCGACCACGGCAAGA-3′ R 5′-AAACGGGTGGACTTGTAGAATTG-3′ PR 5′-FAM-CTA(pdC)ATGGTG(pdC)GTGAGC-ZNA4-BHQ-1-3′
** *β-actin* ** **(control)**	F 5′-GGCACCCAGCACAATGAAG-3′ R 5′-GCCGATCCACACGGAGTACT-3′ PR 5′-Yakima Yellow-TCAAGATCATTGCTCCTCCTGAGCGC-BHQ-1-3′

* Gene ID: http://www.ncbi.nlm.nih.gov/gene (accessed on 1 December 2020). R: Reverse primer, F: Forward primer, PR: Probe. C-pdC: 5-propynyl-dC.

**Table 3 biomedicines-13-03082-t003:** Pairwise comparison of miRNA biogenesis gene expression levels among AML, MDS, and control groups.

Gene	AML- Control (p.adj)	MDS- Control (p.adj)	AML- MDS (p.adj)	Interpretation
*DICER1*	0.028 *	0.019 *	0.642 ns	Downregulated in AML and MDS vs. Control
*DROSHA*	0.031 *	0.025 *	0.708 ns	Decreased in AML and MDS vs. Control
*DGCR8*	0.008 **	0.002 **	0.371 ns	Variable expression; increased trend in AML
*TARBP2*	0.005 **	0.006 **	0.541 ns	Reduced in AML and MDS compared to Control
*AGO1*	0.093 ns	0.078 ns	0.885 ns	No significant difference among groups

Values indicate Bonferroni-adjusted *p*-values (p.adj) (Mann–Whitney U test); *p* < 0.05 (*), *p* < 0.01 (**), ns: not significant.

**Table 4 biomedicines-13-03082-t004:** Diagnostic performance of miRNA-biogenesis genes in MDS and AML compared to controls.

Gene	AUC MDS- Control	AUC AML-Control	*p*	Youden J	Criterion (2^−ΔΔCt^)	Sensitivity (%)	Specificity (%)
** *DGCR8* **	0.960 (0.910–0.990)	0.971 (0.925–0.995)	<0.0001	0.8386	>1.993	98.15	85.71
** *DICER1* **	0.866 (0.810–0.970)	0.950 (0.885–0.995)	<0.0001	0.7646	≤0.835	90.74	85.71
** *DROSHA* **	0.777 (0.690–0.905)	0.886 (0.810–0.960)	<0.0001	0.5847	≤0.915	87.04	71.43
** *TARBP2* **	0.819 (0.650–0.890)	0.875 (0.780–0.950)	0.018	0.6402	≤0.692	92.59	71.43
** *AGO1* **	0.790 (0.680–0.900)	0.836 (0.730–0.930)	<0.0001	0.5370	≤0.831	53.70	100.00

AUC: 95% CI ROC curve. Youden J index and optimal cutoff (criterion) were determined to maximize combined sensitivity and specificity based on 2^−ΔΔCt^ values.

**Table 5 biomedicines-13-03082-t005:** Summary of multigene model validation and calibration performance.

Cohort Comparison	Model AUC	Best Single gene	Best Single Gene AUC	AUC Difference (Model-Best)	DeLong *p*-Value	Brier Score	HL χ^2^ (df)	HL *p*-Value
**MDS-** **Control**	0.978	*DGCR8*	0.964	+0.014	<0.001	0.08	6.12 (8)	0.42
**AML-** **Control**	0.982	*DGCR8*	0.971	+0.011	0.004	0.07	7.04 (8)	0.38

AUC = Area under the ROC Curve; HL = Hosmer-Lemeshow goodness of fit test.

## Data Availability

Due to ethical restrictions and patient confidentiality, individual level data cannot be shared publicly. Aggregated or anonymized data may be provided by the corresponding author upon reasonable request.

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
