# Peer review of "Disrupted miRNA Biogenesis Machinery Reveals Common Molecular Pathways and Diagnostic Potential in MDS and AML"

_biomedicines, 2025, doi:10.3390/biomedicines13123082_

Round 1
Reviewer 1 Report
Comments and Suggestions for Authors
I am really grateful to review this manuscript. In my opinion, this manuscript can be published once some revision is done successfully. I made three suggestions and I would like to ask your kind understanding.
Firstly, it can be noted that the random forest often surpasses logistic regression in terms of prediction power and random forest variable importance and Shapley Additive Explanations (SHAP) dependence plots are very effective to identify the strength and direction of association between disorder (e.g., acute myeloid leukemia AML) and its major predictor such as triglyceride-glucose index (miRNA genesis genes). In this context, I would like to ask the authors to derive random forest variable importance and SHAP dependence plots.
For reference, I would like to introduce brief introductions here. The SHAP value of a predictor for a participant measures the difference between what the random forest predicts for the probability of disorder with and without the predictor. For example, let’s assume that the SHAP values of DICER1 for AML have the range of (0.55, 0.65). Here, some participants have SHAP values as low as 0.55, and other participants have SHAP values as high as 0.65. The inclusion of the predictor (DICER1) into the random forest will increase the probability of the dependent variable (AML) by the range of 0.55 and 0.65. Here, the SHAP dependence plot is expected to (1) show there exists a positive association between the predictor (DICER1) and the dependent variable (AML) and (2) test whether there exists a non-linear association.
Secondly, it can be noted that the area under the curve (AUC) is relatively high but either sensitivity or specificity is relatively low in Table 4. For example, the AUC/sensitivity of DGCR8 for AML is 97%/98% but its specificity counterpart is 86%. On the contrary, the AUC/specificity of AGO1 for AML is 84%/100% but its sensitivity counterpart is 54%. These discrepancies undermine the validity of this study, hence I would like to ask the authors to improve these weaknesses. Thirdly, large language models are garnering great attention now, hence I would like to ask the authors to address their potential applications for the current topic.
Author Response
A detailed point‐by‐point response to all reviewer and editor comments has been prepared and uploaded as a separate file for your evaluation.

Reviewer 2 Report
Comments and Suggestions for Authors
The Authors investigated the role of key miRNA biogenesis genes in the pathogenesis and diagnosis of myelodysplastic syndromes (MDS) and acute myeloid leukemia (AML). Bone marrow samples from newly diagnosed, untreated MDS and AML patients, alongside matched healthy controls, were analyzed for the expression of DROSHA, DGCR8, DICER1, TARBP2, and AGO1 using quantitative real-time PCR. Statistical analyses, including correlation matrices, ROC analyses, multigene logistic modeling, and decision-curve assessment, were performed to explore molecular interactions and evaluate diagnostic utility. The results showed significant downregulation of DROSHA, DICER1, and TARBP2 in MDS and AML, suggesting impaired miRNA maturation and disrupted post-transcriptional regulation, whereas DGCR8 expression increased in higher-risk MDS, indicating compensatory microprocessor activation. AGO1 levels remained stable, implying partial preservation of RISC function. Correlation analyses highlighted a co-regulated DROSHA–TARBP2–AGO1 module, while DGCR8 and DICER1 emerged as the strongest individual diagnostic markers. The integrated five-gene signature achieved high discriminative performance (AUC≈0.98) and demonstrated potential clinical applicability. These findings suggest that coordinated alterations in miRNA biogenesis genes contribute to leukemogenesis in MDS and AML and support the development of miRNA-based diagnostic and therapeutic strategies.
Major concerns:
-
Functional assays are needed to directly link DROSHA, DICER1, and TARBP2 dysregulation to altered hematopoietic differentiation and proliferation.
-
The mechanistic basis of DGCR8 upregulation in high-risk MDS requires further clarification.
-
The stability of AGO1 expression warrants investigation into whether partial RISC activity is sufficient to maintain residual miRNA function.
-
The clinical utility of the five-gene signature should be tested prospectively to evaluate real-world diagnostic performance.
-
Potential confounding factors such as patient age, cytogenetic abnormalities, and comorbidities need to be accounted for in logistic modeling.
-
The Authors should assess whether integrating miRNA expression profiles alongside biogenesis genes improves predictive power.
-
The study lacks longitudinal analyses to determine whether these gene expression patterns predict disease progression or response to therapy.
-
Statistical correction for multiple testing should be explicitly described to ensure robustness of reported correlations and ROC analyses.
-
Detailed methods for normalization and quality control of qPCR data should be provided to ensure reproducibility.
Author Response

(The authors gave the same response as above.)

Reviewer 3 Report
Comments and Suggestions for Authors
The paper investigates the expression of five key genes involved in miRNA biogenesis in bone marrow of patients with MDS and AML, compared to healthy controls. The aim is to determine whether there is a common molecular signature of impaired miRNA processing in these two disorders and whether the expression of these genes may have diagnostic significance.
The study is well-organized and significant for understanding miRNA biogenesis in MDS/AML. The findings are consistent, and the figures are adequate and understandable. However, some limitations reduce the conclusions.
Suggestions for improvement
- The study is based on a relatively small cohort (especially the control), which significantly limits the reliability of the ROC model and comparative conclusions.
- Conclusions on clinical applicability should be more moderate, as an independent validation cohort is lacking.
- ACTB as a reference gene may be variable in hematopoietic tissue, as a result of which 2–ΔΔCt analysis may be problematic, which should be discussed. All of this should be highlighted as a limitation, and therefore, the strength of the claims in the conclusion should be revised.
- Add a table with demographic data and basic laboratory parameters of the control group (if available).
- Explain that FASTQ/miRNA sequencing was not performed and what could be lost with such an approach.
- The results presented for DGCR8 in the context of high-risk MDS are interesting — it is advisable to emphasize their potential biological interpretation in more detail.
- Although the correlation results are well presented, it should be emphasized more clearly that the data do not indicate causality.
- The discussion contains repetitions and overly extensive explanations of mechanisms already mentioned earlier.
Author Response

(The authors gave the same response as above.)

Round 2
Reviewer 1 Report
Comments and Suggestions for Authors
In my opinion, this manuscript can be published in current form.
Author Response
Thank you for your positive evaluation. No further revisions were required.

Reviewer 2 Report
Comments and Suggestions for Authors
The Authors investigated expression patterns of key components of the miRNA biogenesis machinery in bone marrow samples from newly diagnosed and untreated MDS and AML patients compared with matched healthy controls, analyzing DROSHA, DGCR8, DICER1, TARBP2, and AGO1 using quantitative real-time PCR alongside correlation matrices, ROC analysis, and multigene logistic modeling. The Authors found that DROSHA, DICER1, and TARBP2 were markedly reduced in both MDS and AML, DGCR8 expression increased in higher risk MDS, and AGO1 levels were largely preserved, with modeling indicating that DGCR8 and DICER1 were the strongest diagnostic discriminators and that an integrated five-gene signature achieved high diagnostic performance. These findings suggest that coordinated dysregulation of miRNA biogenesis contributes to impaired hematopoiesis and myeloid transformation and may support development of miRNA-based diagnostic and therapeutic strategies.
Major concerns:
-The Authors need to provide detailed information regarding patient demographics, disease subtypes, and sample handling to ensure appropriate comparison across groups.
-They must provide validation of qPCR primer specificity and amplification efficiency for all five genes.
-The Authors should show independent replication of gene expression patterns in an external cohort.
-Data should be provided to demonstrate how potential confounders such as prior inflammatory states or comorbidities were excluded or accounted for.
-Proper assays should be provided to assess whether altered gene expression leads to measurable functional impairment in miRNA processing or downstream hematopoietic pathways.
Author Response
Thank you very much for your detailed and constructive comments. We have carefully revised the manuscript and provided point-by-point responses addressing all recommendations. Your methodological insights substantially improved the clarity, rigor, and scientific quality of the work. A full response document outlining all revisions has been uploaded.

Reviewer 3 Report
Comments and Suggestions for Authors
We thank the authors for their thorough and high-quality revision of the manuscript. The changes you made significantly improved the clarity of the text, the transparency of the methodology, and the balanced interpretation of the results. Key points of concern, previously highlighted - especially the small number of controls, limitations of the ROC model, potential instability of the reference ACTB gene, and lack of functional validation — are now clearly explained, making the manuscript more methodologically and scientifically sound.
The discussion is now better structured, with reduced repetition and stronger rationalization, particularly regarding the role of DGCR8 in the progression of high-risk MDS and the functional implications of TARBP2 in AML. The abstract has also been significantly improved, now more accurately reflecting the study's strengths and limitations. We also appreciate the additions to the methods, including a clearer description of primer design, statistical steps, and constraints related to control sampling.
Author Response
Thank you very much for your constructive and positive feedback. A detailed response has been uploaded in the reviewer reply file.

Round 3
Reviewer 2 Report
Comments and Suggestions for Authors
-
Author Response
Response to Reviewer
We sincerely thank the Reviewer for the time and effort devoted to evaluating our manuscript and for the constructive comments provided in the previous round. All major concerns raised during Round 2 have been carefully addressed in the revised version of the manuscript.
In accordance with the Reviewer’s suggestions, we have implemented comprehensive improvements to the English language throughout the manuscript and refined several sentences for clarity and precision. Additionally, all figures have been re-examined and enhanced to ensure high-resolution quality and improved readability, fully in line with the Reviewer’s earlier recommendations.
We appreciate the Reviewer’s valuable contribution to strengthening our manuscript and remain grateful for the thoughtful feedback provided during the review process.
